# Clinical Utility of Quantitative HBV Core Antibodies for Solving Diagnostic Dilemmas

**DOI:** 10.3390/v15020373

**Published:** 2023-01-28

**Authors:** Ivana Lazarevic, Ana Banko, Danijela Miljanovic, Maja Cupic

**Affiliations:** Institute of Microbiology and Immunology, Faculty of Medicine, University of Belgrade, 11000 Belgrade, Serbia

**Keywords:** hepatitis B virus (HBV), HBV core antibody, quantitation, qAnti-HBc, diagnostic method, new biomarker, clinical utility

## Abstract

The present-day management of hepatitis B virus (HBV) infection relies on constant and appropriate monitoring of viral activity, disease progression and treatment response. Traditional HBV infection biomarkers have many limitations in predicting clinical outcomes or therapy success. Quantitation of HBV core antibodies (qAnti-HBc) is a new non-invasive biomarker that can be used in solving multiple diagnostic problems. It was shown to correlate well with infection phases, level of hepatic inflammation and fibrosis, exacerbations during chronic infection and presence of occult infection. Further, the level of qAnti-HBc was recognised as predictive of spontaneous or therapy-induced HBeAg and HBsAg seroclearance, relapse after therapy discontinuation, re-infection after liver transplantation and viral reactivation upon immunosuppression. However, qAnti-HBc cannot be relied upon as a single diagnostic test to solve all dilemmas, and its diagnostic and prognostic power can be much improved when combined with other diagnostic biomarkers (HBV DNA, HBeAg, qHBsAg and anti-HBs antibodies). The availability of commercial qAnti-HBc diagnostic kits still needs to be improved. The comparison of results from different studies and definitions of universal cut-off values continue to be hindered because many methods are only semi-quantitative. The clinical utility of qAnti-HBc and the methods used for its measurement are the focus of this review.

## 1. Introduction

Almost two-billion people worldwide have been infected with the hepatitis B virus (HBV) at some point in their lives, and, according to World Health Organization’s (WHO) estimation, 296-million people live with chronic infection [1,2]. Despite successful vaccination programs and the decline of HBsAg seroprevalence since 2000, HBV remains a global health problem since complications related to chronic infection are still a source of significant morbidity and mortality. Approximately 820,000 deaths yearly occur from HBV-induced liver cirrhosis and hepatocellular carcinoma (HCC) [2,3].

HBV infection develops as either acute or chronic and can lead to a broad spectrum of clinical manifestations. Chronic HBV infection is a dynamic process resulting from the complex interaction between the virus and the host’s immune response but not all chronically infected patients have chronic hepatitis B (CHB). According to the European Association for the Study of the Liver (EASL), the natural history of chronic HBV infection progresses through 5 distinct but not necessarily successive phases (Figure 1): (1) HBeAg-positive chronic HBV infection (‘‘immune tolerant’’ phase) is characterised by detectable serum HBeAg, high HBV DNA levels, normal alanine aminotransferase (ALT) levels and minimal or no liver necroinflammation or fibrosis; (2) HBeAg-positive CHB (“immune-active” phase) represents active inflammatory disease with positive HBeAg, high HBV DNA levels, elevated ALT levels and moderate or severe liver necroinflammation and accelerated progression of fibrosis; (3) HBeAg-negative chronic HBV infection (“immune-control” or “inactive carrier” phase) occurs after seroconversion of HBeAg-positive to HBeAg-negative/anti-HBe-positive state. HBV DNA is usually low (<2000 IU/mL) or undetectable, while ALT levels are normal and the HBsAg level is typically low (<1000 IU/mL). There is a minimal necroinflammatory activity in the liver, mild fibrosis and low risk of progression to cirrhosis; (4) HBeAg-negative active chronic hepatitis (“immune escape-mutant” phase) follows in some patients and is marked by an HBeAg-negative/anti-HBe-positive state, moderate to high levels of HBV DNA (>2000 IU/mL) and fluctuating or elevated ALT levels. There is considerable hepatic necroinflammatory activity and rapid progression to cirrhosis. In most cases, HBeAg expression in this phase is reduced or abolished as a result of mutations present in pre-core (Pre-C) and/or basal core promoter (BCP) regions; (5) HBsAg-negative phase (‘‘occult HBV infection”) may occur in some patients and is defined by absent serum HBsAg and presence of anti-HBc antibodies with or without anti-HBs antibodies. The serum DNA may or may not be detectable, but covalently closed circular DNA (cccDNA) is present in the liver. Patients who undergo immunosuppression in this phase may experience reactivation of HBV infection. The occult infection should be differentiated from the state of the “functional cure” after antiviral therapy. In both cases, HBsAg and HBV DNA are undetectable while cccDNA is present in the liver. Yet in the “functional cure,” there is a complete suppression of viral replication associated with normal liver function and a low chance of fibrosis progression [4]. According to hepatic necroinflammatory activity, risks of liver disease progression and presence of specific biomarkers, phases 2 and 4 are sometimes referred to as “hepatitis phases”, while 1, 3 and 5 are defined as “infection phases”.

The present-day management of chronic HBV infection relies on constant and appropriate monitoring of viral activity, disease progression and treatment response. Traditional HBV infection biomarkers, such as HBV DNA, hepatitis B surface antigen (HBsAg), hepatitis B e antigen (HBeAg), antibodies to hepatitis B core, e and surface antigens (anti-HBc, anti-HBe and anti-HBs), have been shown to have many limitations in predicting clinical outcome or therapy success [5]. Detection and quantitation of viral cccDNA in liver cells is the gold standard for obtaining information on HBV replicative and transcriptional activity [6,7]. However, it is unlikely that cccDNA would become a routine biomarker because liver biopsy is required for its analysis, and there still needs to be standardised methods for its quantitation. Additional methods are required for assessing cccDNA transcriptional activity, which can be present at different levels or entirely absent. On the other hand, HBsAg presence is not dependent on cccDNA transcription alone, but its main sources are the subviral particles derived from viral DNA randomly integrated into the host genome [6,8]. Thus, an urgent need has been developed for new non-invasive biomarkers capable of reflecting the intrahepatic activity of the virus and assessing the risk of liver-related complications and the likelihood of achieving therapy endpoints [8]. Some of these new biomarkers include quantitative HBsAg (qHBsAg), serum HBV RNA, hepatitis B core-related antigen (HBcrAg), quantitative HBeAg (qHBeAg) and quantitative anti-HBc (qAnti-HBc) antibodies [6,7,9]. The clinical utility of HBV core antibody quantitation and the methods used for their measurement are the focus of this review.

## 2. HBV Core Antibodies as a Classical Diagnostic Marker of HBV Infection

Hepatitis B core protein originates from the nucleocapsid that encapsulates viral DNA. Pre-core/core precursor molecule is needed to produce both core protein and HBeAg. The production of HBeAg requires the maturation of the precursor molecule in the endoplasmic reticulum, after which it is secreted as a soluble antigen and serves as a valuable serological marker. Both antigens (HBcAg and HBeAg), together with 22-kDa truncated pre-core protein, constitute HBcrAg, a new biomarker of intrahepatic HBV cccDNA and its transcription [6].

HBcAg epitopes are very potent immunogens, able to induce a cellular and humoral immune response, manifested in T-cell proliferation and production of anti-HBc antibodies during natural HBV infection [10].

Anti-HBc is a classical serological marker comprising anti-HBc IgM and IgG antibodies. While anti-HBc IgM antibodies are present during active liver inflammation and disappear in the recovery stage, anti-HBc IgG can persist for a long time, regardless of an ongoing infection or virus clearance in an HBV-infected host [11,12]. Thus, anti-HBc IgG is a broadly used marker to identify individuals who were ever infected with HBV, whereas IgM anti-HBc is used to identify acute infection. Total anti-HBc can be found in serum very soon after detection of HBsAg and can persist continuously for 10–20 years or, very often, life-long. It is the only marker present in all phases of chronic HBV infection and is positive in nearly 100% of chronically infected patients and 80–99% of individuals with occult infection [13]. However, anti-HBc antibodies can be permanently or intermittently absent in HBV-infected immunocompromised individuals [14].

Over a decade ago, an idea was born that quantitation of anti-HBc (total, IgG or IgM) can bring further information on the status of immune activation in an infected individual and thus provide additional understanding of the phase of infection, stage of liver disease and sensitivity to treatment [15]. The level of HBV core antibodies was expected to be a surrogate marker for both the activity of HBV-specific adaptive immune response and the intrahepatic HBcAg load. 

## 3. Methods and Units for Anti-HBc Antibodies Quantitation

An international reference standard has been developed for HBV core antibodies quantitation, measured in WHO IU/mL [16]. The International Standard (NIBSC 951522) is assigned a unit of 50 IU/mL and is equivalent to the 100 PEI-U/mL of the previous Paul-Erlich-Institute anti-HBc standard (PEI 82). This standard is designed to calibrate anti-HBc kits’ sensitivity, secondary standards, and quality control procedures. Nonetheless, more than one currently available method is only semi-quantitative, expressing qAnti-HBc in units other than IU/mL and impeding comparing results between different studies.

Currently used anti-HBc assays are based on competitive/inhibitory, indirect or double-antigen sandwich immunoassay technology and are developed as enzyme-linked immunosorbent assay (ELISA), chemiluminescent microparticle immunoassay (CMIA), chemiluminescent immunoassay (CLIA) or chemiluminescent enzyme immunoassay (CLEIA). The methods and units used for anti-HBc antibodies quantitation are summarised in Table 1.

The double-antigen sandwich immunoassay for qAnti-HBc was first designed in 2010 and is reported to be superior to others in sensitivity and specificity, although with a relatively narrow quantitation range [15,17,18]. ELISA based on the double-antigen sandwich immunoassay is now commercially available (Wantai Biological Pharmacy Enterprise Company, Beijing, China) and can be calibrated against WHO International Standard for results to be reported as IU/mL. The linear detection range was reported to be 2–5 log IU/mL, and the lower limit of quantitation (LLoQ) was 0.25 IU/mL [19]. 

Another frequently used assay is Lumipulse G HBcAb-N, a fully automated two-step sandwich CLEIA (Fujirebio, Tokyo, Japan) [20,21]. In this method, anti-HBc IgG levels are automatically reported as the cut-off index (COI), calculated as a multiple of the cut-off value obtained from calibration data (COI = S/C × 0.09). The manufacturer reported the lower limit of quantitation to be 1 COI. The results, however, can be calibrated against WHO International Standard (NIBSC 951522) to obtain IU/mL and the reported lower limit of detection (LLoD) and LLoQ were 0.5 and 0.8, respectively [20]. Compared to other qAnti-HBc methods using chemiluminescence, this one showed the highest sensitivity, specificity and the broadest linear dynamic range [22]. 

Several CMIA methods are utilised for anti-HBc quantitation: Architect System HBcAb II for total anti-HBc, anti-HBc IgM (Abbott Diagnostics, Lake Forest, IL, USA) and anti-HBc QN (InnoDx Biotech, Xiamen, China). CMIA methods developed by Abbott are not designed for quantitation, but the results are expressed as the ratio of S(sample)/Co(cut-off) of relative light units/mL (RLU/mL), where the result ≥1 is defined as positive. These results were converted and reported in PEI U/mL in some studies [23]. The manufacturer informs the method’s sensitivity to be 0.4–0.5 PEI U/mL. 

A commercial CLIA method for anti-HBc quantitation (Autobio Diagnostics, Zhengzhou, China) reports results in NCU/mL (China National Clinical Units). As presented in a recent study, it could be calibrated using the PEI standard and expressed as PEI U/mL [24].

In addition, a first, simple and rapid fluorescence point-of-care (POC) test based on a lateral flow immunoassay (LFIA) method has been developed for the determination of anti-HBc concentrations in serum [25]. It is a competitive time-resolved fluoroimmunoassay (TRF-IA) that uses carboxylate-modified polystyrene Eu (III) chelate microparticles as a reporter and a portable TRF strip reader for measuring fluorescence. The assay is reported to have a very low detection limit; it is also reported to be very rapid (15 min) and inexpensive.

**Table 1 viruses-15-00373-t001:** Methods and units used for anti-HBc antibodies quantitation.

Method	Manufacturer	Unit	Reference
Double-antigen sandwich ELISA	Wantai Biological Pharmacy Enterprise Company, Beijing, China	IU/mL	[19]
Two-step sandwich CLEIA	Fujirebio, Tokyo, Japan	Cut-off index (COI)	[20]
CMIA for total anti-HBc and anti-HBc IgM	Abbott Diagnostics, IL, USA	S (sample)/Co (cut-off) of RLU/mL	[23]
CMIA	InnoDx Biotech, Xiamen, China	IU/mL	[26]
CLIA	Autobio Diagnostics, Zhengzhou, China	NCU/mL	[24]
Lateral flow immunoassay (LFIA)	Non-commercial	IU/mL	[25]

ELISA—enzyme-linked immunosorbent assay; CLEIA—chemiluminescent enzyme immunoassay; CMIA—chemiluminescent microparticle immunoassay; CLIA—chemiluminescent immunoassay; IU—international units; RLU—relative light units; NCU—China National Clinical Units.

## 4. Clinical Utility of qAnti-HBc

The clinical utility of HBV core antibody quantitation is widely investigated by determining its value in solving different diagnostic dilemmas. The significant findings, together with proposed cut-off values and relevant references, are summarised in Figure 2.

### 4.1. Differentiation of Phases of CHB Infection

The efficacy of anti-HBc measurement for monitoring the natural history of HBV was one of the first to be investigated upon establishing the method. The main idea was that insight into changes in anti-HBc levels throughout phases of chronic infection can expand knowledge of the pathogenesis and immune activation during infection. The studies designed to investigate this included treatment-naïve individuals in different phases of chronic infection, some of them including individuals having an occult infection or evidence of past infection. Two initial studies conducted in 2014 showed the levels of core antibodies to be significantly lower (approximately 1000-fold) in individuals who were HBsAg negative, i.e., those with occult or resolved infection than in chronically infected individuals (with persistent HBsAg positivity) [27,28]. The results of both studies showed variations of qAnti-HBc mean values between HBsAg-positive phases of infection. The levels were considerably different, approximately 10-fold higher in immune active phases 2 and 4 than in immune tolerant or inactive phases 1 and 3. Nonetheless, no significant difference in mean values was observed between “low level” phases 1 and 3 or “high level” phases 2 and 4, making this marker unsuitable for differentiation. Similarly, in a 2018 study, the anti-HBc levels increased from phase 1 to phase 2, then decreased and raised again from phase 3 to phase 4 [18]. Phase 4 was the one that showed a significantly higher mean level than the other three. The phase with the lowest mean value of anti-HBc was phase 1 in all three studies, unlike a more recent study from 2021, which reported a very similar pattern of anti-HBc variations but with the lowest level in inactive phase 3 [29]. On the other hand, the latest report showed a considerably different way of anti-HBc level changes through phases of infection [26]. Mean levels constantly increased, very sharply from phases 1 to 2, then moderately from 2 to 3 and again sharply from 3 to 4, where the mean values were highest among all phases. The qAnti-HBc level was higher than the lower limit of detection (LLoD) (2 log IU/mL) in all phases, while the frequency of anti-HBc level above the upper limit of detection (5 log IU/mL) was similar in HBeAg-positive phases, 1 and 2, and significantly higher in active (4) than in inactive (3) HBeAg-negative phases.

Anti-HBc antibodies are the only marker present in all phases of the natural history of HBV infection, including around 90% of occult infection cases. This is based on the outstanding immunogenic potential of HBV capsid, and it can be assumed that the level of core antibodies depends on the degree of exposure of this antigen to immune system cells [72]. Thus, an increased presence of core antigen is the reason for the higher production of antibodies during phases of overt infection than in the occult phase or the past infection. However, this correlation needs to be clarified when core antibody levels are analysed in different phases of chronic infection. Higher anti-HBc levels are observed in phases of active hepatic necroinflammatory activity (2 and 4), which implies its dependence more on immune status and hepatitis activity than on the level of HBV DNA or the presence of core antigen [27].

The active genome replication and transcription of antigens should be stimulating for anti-HBc synthesis, but the release of HBcAg as part of the virions may be less potent than a release of HBcAg after cell lysis. Core antigen hidden under the envelope in virions is not so reachable for B lymphocytes which can explain lower levels of anti-HBc in the initial “immune tolerant” phase than in the “immune active” phase. The dynamic of HBcAg expression and its secretion in the serum has yet to be entirely clarified. Still, it is speculated that immune cells are exposed to the immunogenic property of this antigen during cell lysis since numerous studies report a simultaneous increase of anti-HBc antibodies with ALT and aspartate aminotransferase (AST) [18,20,27,28,29]. ALT was used as a surrogate marker of the immune response during CHB because ALT levels were associated with T-cell mediated hepatocyte lysis but had poor specificity since it can be affected by drug and alcohol abuse, fatigue and infection with other hepatotropic viruses [56]. It was observed that the elevation of ALT can be preceded by a peak of anti-HBc in some individuals [27]. In cases when ALT is normal, during phase 1 or after therapy, anti-HBc can be more beneficial for assessing the host immune response. Together with AST, qAnti-HBc can more accurately identify the immune tolerant phase in HBeAg-positive patients with normal ALT and high viral loads [30]. 

Mutations are frequently observed in the HBV genome during chronic infection due to HBV reverse transcriptase’s lack of proofreading activity. Basal core promoter (BCP) and pre-core (Pre-C) mutations are the cause of reduced or abolished HBeAg expression and are commonly associated with phase 4 of chronic infection (“immune escape-mutant” phase). However, these mutations were shown to exist in 16% of patients during phase 1, and their presence was associated with qAnti-HBc level [24]. Patients infected with BCP/Pre-C mutant strains had higher qAnti-HBc levels than patients infected with wild-type strains during phase 1 of CHB. These findings may be significant because the qAnti-HBc level can assist in recognising at-risk patients in the early stages of CHB, as these mutations are linked with faster progression of liver injury and risk of HCC development [73].

In addition, qAnti-HBc was proven to be helpful in discriminating HBeAg-negative phases, i.e., inactive carriers from patients with active HBeAg-negative hepatitis [64]. While qAnti-HBc level in HBeAg-positive phases is associated with ALT but not HBV DNA, thus reflecting the strength of the immune response, in HBeAg-negative phases, it is determined by intrahepatic HBcAg load, which correlates with cccDNA [33].

### 4.2. Prediction of Spontaneous HBeAg Seroclearance

In patients with chronic HBV infection, HBeAg seroclearance indicates a transition from phase 2 (“immune active”) to phase 3 (“inactive carrier”) with a lower level of HBV DNA replication and reduced risk of liver disease progression. Predictors of HBeAg seroclearance to treat naïve patients of an older age, the lower level of HBV DNA, higher level of ALT, presence of Pre-C/BCP mutations and viral genotype [74,75]. Since the benefits of antiviral treatment in HBeAg-positive patients are still debatable, recognising the probability of spontaneous HBeAg loss may help weigh the risks and benefits of initiating therapy. There were some suggestions that new HBV biomarkers like HBcrAg, HBV RNA and qAnti-HBc might predict HBeAg loss [6,7]. A high level of qAnti-HBc was proven to be an independent predictor of HBeAg loss in untreated adults, unlike HBcrAg [31]. Together with a lower level of HBsAg and viral genotype B and B + C, a higher qAnti-HBc level (>2.7 log IU/mL) was demonstrated to be predictive of spontaneous HBeAg seroconversion in children with normal ALT [32].

### 4.3. Prediction of Spontaneous HBsAg Seroclearance

The HBeAg-negative state during chronic HBV infection is associated with a broad spectrum of clinical conditions ranging from the inactive carrier with an overall survival comparable to HBV non-infected individuals to active chronic hepatitis with considerable hepatic necroinflammatory activity and rapid progression to cirrhosis [4]. There is always an unpredictable transition risk from an inactive carrier state to an active infection. Current antiviral treatment can alter the course of active infection by stopping fibrosis progression and reducing the chances of end-stage complications. Thus, it is essential to distinguish actual inactive carriers from patients with constant low viremic active infection (HBV DNA >2000 IU/mL with normal ALT) and to predict the possibility of HBV DNA undetectability and HBsAg seroclearance. Unlike in HBeAg-positive phases, the more favourable clinical outcome in HBeAg-negative phases is associated with a lower level of qAnti-HBc [33,34].

Patients in actual inactive carrier state were proven to have a significantly lower level of qAnti-HBc than HBe-Ag negative patients with constant low viremia (>2000 IU/mL) and normal ALT [34]. Further, 90% of patients with actual active HBeAg-negative infection had qAnti-HBc above 4.22 IU/mL, in contrast to only 16.5% of low-viremic HBeAg-negative patients. The diagnostic accuracy for identifying inactive carrier profiles was improved when three biomarkers were employed—HBV DNA level, HBcrAg and qAnti-HBc. The authors of this study concluded that HBeAg-negative infection with low viremia was associated with benign outcomes for most of these patients were bound to transition to inactive carrier profile, which in 20% of cases was a prelude of HBsAg seroclearance within five years.

HBsAg seroclearance represents the host’s immunity against HBV and a favourable clinical outcome. Lower baseline HBsAg level, lower level of HBV DNA and inactive carrier state are proven predictors of HBsAg loss [75,76]. Nonetheless, the course of the disease can still be unpredictable, creating a demand for additional biomarkers. Baseline anti-HBc level <3 log IU/mL (sensitivity 30.4% and specificity 91.7%) was associated with spontaneous HBV DNA undetectability and HBsAg seroclearance in treatment-naïve HBeAg-seronegative CHB patients [33]. The probability of this outcome was further enhanced by adding HBsAg level <2 log IU/mL. In addition, the qAnti-HBc level decreased consistently over a time before a favourable outcome. Since qHBsAg level originates not only from viral cccDNA but also from HBV DNA integrated into the host’s genome, qAnti-HBc relating to HBcAg load in the liver can better reflect the residual viral cccDNA in low viremic HBeAg-negative phases.

### 4.4. Diagnosis of Acute Exacerbations during CHB

Phases of active CHB are marked with intermittent acute exacerbations and episodes of normal liver function. In HBV endemic areas, acute exacerbations of CHB are common and often represent the first sign of disease. Clinical and serological evidence of acute exacerbations in CHB is very similar to those in acute hepatitis B, which is why differentiation between these two clinical entities presents a considerable diagnostic dilemma. Anti-HBc IgM has been considered the most critical marker for recognising acute HBV infection [77]. However, modern diagnostic methods with higher sensitivity revealed a considerable percentage of anti-HBc IgM-positive cases among patients with CHB but with lower quantitative levels [78,79]. Lull et al. demonstrated that higher levels of qAnti-HBc IgM and lower INR helped establish the diagnosis of acute infection [35]. Further, it was concluded that only IgM qAnti-HBc was a significant discriminating factor between acute exacerbation of CHB and acute infection and that combining it with other markers did not add to its discriminatory power.

Acute-on-chronic liver failure (ACLF) is a clinical syndrome manifesting as acute deterioration of pre-existing liver disease following a precipitating event, and it is associated with high short-term mortality because of multisystem organ failure [80]. It is a standard model of end-stage liver disease in patients with chronic HBV infection. Since the host’s hyper-inflammatory status seems responsible for multisystem organ failure and high mortality, an immune-related marker is expected to be a good choice for clinical prognosis. The first study that investigated the performance of qAnti-HBc level in patients with ACLF demonstrated that qAnti-HBc level was significantly higher in patients with ACLF than in patients with CHB regardless of the HBeAg status [36]. On the other hand, the qAnti-HBc level was notably lower in patients with the fatal outcome than in patients who survived ACLF. Indeed, high qAnti-HBc level (≥5 log IU/mL) was an independent factor associated with survival in patients with HBV DNA <5 log IU/mL. The significant level of qAnti-HBc could reflect hyper-inflammatory status associated with ACLF, while the lower level of qAnti-HBc in fatal outcomes could result from overall immunological anergy.

### 4.5. Diagnosis of Occult Infection and Prediction of Viral Reactivation

Occult HBV infection (OBI) is defined as the presence of replication-competent HBV DNA (mostly episomal cccDNA) in the liver with or without the presence of HBV DNA in the serum of individuals who test negative for hepatitis B surface antigen (HBsAg) by currently available assays [13]. Detectability of HBV DNA in serum is usually intermittent and frequently in the low viral load range (<200 IU/mL). OBI is classified as seropositive when anti-HBc and/or anti-HBs are present and less common seronegative when HBV DNA (mostly intrahepatic) is the only available marker [81]. Seropositive OBI results from HBsAg loss either after recovery from acute infection or during long-lasting chronic infection, while seronegative emerges from progressive loss of antibodies or their absence from the beginning. The molecular basis of OBI lies in the stability and persistence of cccDNA in the hepatocytes. The undetectability of HBsAg is ascribed to the suppression of viral replication and antigen expression, which arise from either epigenetic mechanisms or the host’s immune control. Only in rare cases OBI results from an infection with S gene mutants, able to produce modified HBsAg that can lead to the failure of immunoassays for HBsAg recognition. This type of OBI is associated with the level of viral replication corresponding to those in overt infection. 

Accurate OBI diagnosis requires a liver biopsy and detection of intrahepatic HBV DNA. However, OBI is an asymptomatic condition, and since it is unethical to submit healthy individuals to an invasive procedure, diagnosis is often dependent upon the sensitivity of HBsAg and HBV DNA assays. Although the definition of OBI is not limited to an isolated anti-HBc pattern, detecting anti-HBc is often the first step in establishing the diagnosis of seropositive OBI [82]. There is growing evidence that quantitation of anti-HBc can distinguish OBI from other conditions with positive anti-HBc. As previously mentioned, the levels of qAnti-HBc were shown to be significantly lower (approximately 1000-fold) in HBsAg-negative infections (occult or resolved) than in HBsAg-positive chronic infections [20,27,28]. Further, the average qAnti-HBc level was significantly higher in occult than in resolved infections (approximately 4-fold), and the cut-off for recognition of occult infection was set at >6.6 IU/mL (sensitivity 60.7% and specificity 75.3%) [27]. This was also established in a rare study including liver specimens collected at grafting from anti-HBc-positive liver donors, where total intrahepatic HBV DNA and cccDNA were examined in individuals without liver disease and compared with serum markers [37]. Although no statistical difference was found, the qAnti-HBc level was higher in OBI-positive individuals (total intrahepatic HBV DNA positive) with than without detectible cccDNA. It was demonstrated that qAnti-HBc level above 4.4 COI (sensitivity 92.6% and specificity 48%) was associated with HBV cccDNA detection and thus possible risk of reactivation. Surprisingly, qAnti-HBc level correlated better than HBcrAg with cccDNA. 

Clinical implications of OBI include the further progression of liver disease and the risk of HCC development, possible HBV transmission by blood transfusion or organ transplantation and HBV reactivation upon immunosuppression [81]. Many countries have established testing for anti-HBc in all blood donors to recognise donors with occult infection. In endemic regions with a high prevalence of anti-HBc, the exclusion of donors positive for this marker would result in a severe blood supply shortage [38]. Highly sensitive nucleic acid testing (NAT) is used in blood donation screening, but it is often unavailable in low-income countries, and its sensitivity significantly decreases when applied to mini-pools of multiple donations [81]. Thus, given the previously mentioned findings of a higher level of qAnti-HBc in the occult than resolved infection, qAnti-HBc testing may appear as an inexpensive and convenient biomarker to complement NAT [27,38]. 

Reactivation of HBV infection is defined as the abrupt reappearance of HBV in the serum of a person with previously resolved infection or a marked increase of HBV replication (>2 log from baseline level) in an immunosuppressed patient with once stable chronic infection [83]. The again established significant viral replication may lead to severe hepatitis flare and liver failure. HBsAg-positive patients are eight times more likely to experience reactivation than HBsAg negative/anti-HBc positive patients, but about 40% of OBI patients undergoing immunosuppressive therapy or cancer chemotherapy would also encounter reactivation [13,84,85]. The risk is exceptionally high in patients receiving regimens with anti-CD20 antibodies or undergoing haematological stem cell transplantation (HSCT) [86]. There is a notion supported by evidence from some studies that qAnti-HBc level can be a valuable marker for identifying OBI patients with a higher level of replication-competent cccDNA and, thus, a more increased risk of hepatitis reactivation upon immunosuppression. The authors from Asia demonstrated that patients receiving chemotherapy for lymphoma, with high anti-HBc (>6.41 IU/mL) and low anti-HBs (<56.48 mIU/mL) at baseline, tended to have more chances for development of reactivation and more frequent and severe hepatitis flare [39]. This observation was confirmed by a more recent study involving European patients undergoing HSCT, where reactivation risk (and its prediction) was associated with qAnti-HBc levels >3 COI and persistent or declining anti-HBs levels (<50 mIU/mL) [21]. This leads to a conclusion that baseline qAnti-HBc level is related to the residual HBcAg produced by the reservoir of replication-competent cccDNA and also that transcriptional activity of cccDNA is under immune response control (particularly anti-HBs).

Later during reactivation, the increase of qAnti-HBc level was found to follow the rise in HBV DNA level, possibly associated with the increment in HBcAg production and/or release from damaged hepatocytes [39]. A strong correlation between qAnti-HBc serum levels and HBV cccDNA values was confirmed in other studies [20,37]. As mentioned previously, in immunocompetent individuals with HBsAg-negative status, qAnti-HBc level of >4.4 COI was linked with the possibility of cccDNA detection. In contrast, in HIV-infected individuals, >15 COI qAnti-HBc was related to cryptic HBV DNA replication (not detected by standard quantitative DNA assays) [37,40]. The existence of a non-invasive serum biomarker, able to reflect the cccDNA pool and host’s immune response, can significantly assist in defining endpoints for novel therapies and prophylactic strategies for patients at risk of HBV reactivation. Determining the suitable antiviral prophylaxis protocol for patients at risk for reactivation is still challenging in terms of appropriate choice of drug and duration, mainly because it is often recorded after prophylaxis cessation.

### 4.6. Prediction of Hepatic Inflammation Grade and Stage of Fibrosis 

Active hepatic inflammation is the principal risk factor for developing severe fibrosis, cirrhosis, and hepatocellular carcinoma. Thus, the priority in assessing patients with CHB is evaluating the initial grade of liver inflammation and its further progression. According to guidelines, early control of liver inflammation development is essential to improve the long-term prognosis of these patients [4,87].

Histological analysis of liver biopsy specimens has been the “gold standard” for estimating hepatic inflammation grade. The invasive nature of this procedure, risks of complications, high cost and possible sampling errors have limited the use of percutaneous liver biopsy and initiated the search for reliable non-invasive markers of liver inflammation. In clinical practice, serum ALT has been considered an easily accessible surrogate marker for monitoring the state of liver inflammation. However, it has been shown in multiple studies that nearly half of patients with normal ALT levels had significant liver inflammation [41,42,88,89]. Thus, relying on ALT alone may lead to an underestimation of the proportion of patients who need antiviral therapy. The precision of ALT as a predictive marker is further compromised by not having a uniformly defined upper limit of normal (ULN) among different countries [43].

There is mounting evidence that qAnti-HBc level not only correlates positively with the severity of liver inflammation but also decreases with the improvement of inflammation severity in patients receiving therapy [19,41,42,43,44,45,46,47,48]. It was shown that serum qAnti-HBc level, ALT and liver inflammation activity were closely related [44,47]. The background of this association is the interplay between the virus and the host’s immune system. Since HBV does not have a direct cytopathic effect on hepatocytes, liver tissue damage is, by all evidence, the result of the immune system’s attack on virus-harbouring cells. Both ALT and HBcAg are released upon destruction of hepatocytes and are responsible for the stimulation of B-cells and increase of anti-HBc level. Serum qAnti-HBc levels were proven to be a promising biomarker for the prediction of moderate-to-severe hepatic inflammation, and the accuracy of prediction was improved by combining qAnti-HBc with ALT [44,47]. A serum qAnti-HBc cut-off value of 4.36 log IU/mL provided a sensitivity of 71.69% and specificity of 73.81% in HBeAg-positive patients for prediction of ≥G2 liver inflammation, while a cut-off value of 4.62 log IU/mL with a sensitivity of 54.29% and specificity of 90% was predictive for ≥G2 in HBeAg-negative patients [47]. However, the correlation with ALT ceased to exist when ALT levels were >5 × ULN, probably because the immune system could only be activated to a certain level [27,28,47]. The diagnostic efficiency of qAnti-HBc and ALT was reported to be further improved by adding AST and cytokine CXCL11 [44]. 

While some studies report that qAnti-HBc levels were helpful in the prediction of moderate-to-severe inflammation in both HBeAg-positive and HBeAg-negative CHB phases, others indicate that this benefit is more exhibited in HBe-positive patients [19,44,47]. Most authors emphasise that qAnti-HBc alone is not enough to predict the inflammation grade accurately and that its combination with other biomarkers is desirable [44,45,47,49]. There is a conclusion from a recent study that combined qAnti-HBc, ALT, and HBsAg offer better performance in predicting significant liver inflammation than qAnti-HBc alone or in combination with only ALT [49]. According to a recent study, following the worsening of inflammation and progression of fibrosis through phases of CHB, qAnti-HBc levels were increasing slowly but constantly from HBeAg-positive to HBeAg-negative phases. They were thus deemed not valuable enough for monitoring stages of liver injury [45]. The value of qAnti-HBc in predicting the inflammation level of HBeAg-positive patients was presented to be close to that of qHBsAg and superior to HBV DNA, while in HBeAg-negative patients, it was inferior to HBV DNA but better than qHBsAg.

According to current guidelines, no therapy is recommended for CHB patients with ALT ≤ULN unless significant necroinflammation and/or fibrosis are present [4,87,90]. Nevertheless, patients with ALT ≤ULN, either in phase 1 or 3, frequently display significant liver inflammation or fibrosis. When combined patients from the “immune tolerant” and “inactive carrier” phases with normal ALT levels were included, a severe level of liver injury was found in 78%, while in inactive carriers only, around 35–45% [41,42,43,48]. Thus, there is a growing need for new non-invasive biomarkers in patients with normal ALT levels for assessment of the liver injury stage to recognise candidates for urgent therapy. Among patients with normal ALT levels and detectable DNA, those with moderate to severe liver inflammation (G2–G4, according to the Scheuer classification) had significantly higher qAnti-HBc levels than patients with no or mild inflammation (G0–G1) [41]. Further, the high qAnti-HBc level was discovered to be the only independent predictor of moderate to severe liver inflammation in patients with normal ALT [41,42].

The repeated cycles of hepatocyte damage and repair during phases of active liver inflammation involve recurrent deposition of extracellular matrix, bringing to the progression of liver fibrosis and, ultimately, the development of cirrhosis [91]. HBcAgs, released upon destruction of hepatocytes, are responsible for stimulating B-cells, but there is evidence that anti-HBc secreting B-lymphocytes have hepatotoxic effects and play a vital role in the progression and severity of chronic HBV infection [92]. The assessment of liver fibrosis is essential for disease prognosis and decision-making concerning therapy and serves as crucial proof of response to treatment. Presently, several non-invasive methods exist for evaluating fibrosis stages, but their diagnostic accuracy is not perfect and is sometimes valuable only to exclude severe fibrosis or cirrhosis. Quantitative anti-HBc was shown to help screen for significant fibrosis in treatment-naïve patients [50,51]. Significantly higher qAnti-HBc levels were found in both HBeAg-positive and HBeAg-negative patients with moderate to severe fibrosis (S2–S4, according to the Scheuer classification) than in patients with no or mild fibrosis (S0–S1). The difference in qAnti-HBc levels could not be observed between S3 and S4, and the possible explanation for this inaccuracy was a decrease in viral replication (and HBcAg production) in still-viable hepatocytes during the late stages of fibrosis. The authors also proposed a cut-off value for recognising patients with significant fibrosis (S ≥ 2) to be >4.58 log IU/mL (sensitivity 63.08% and specificity 74.83%) for HBeAg-positive and >4.37 log IU/mL (sensitivity 75.53% and specificity 56.1%) for HBeAg-negative patients. Combining qAnti-HBc with two routine serum parameters, PLT (platelet count) and A/G (albumin to globulin ratio), achieved further improvement of the diagnostic accuracy [52]. On the other hand, Cruchet et al. (2020) did not observe an association of baseline qAnti-HBc level with liver fibrosis evolution in tenofovir-treated HIV/HBV co-infected patients [53]. Curiously, the increase of qAnti-HBc level was linked with a transition to lower fibrosis levels in these patients.

### 4.7. Prediction of Response after Therapy for HCC and Liver Transplantation

It is known today that HBV infection can promote liver disease progression through various mechanisms, ultimately leading to the malignant transformation of hepatocytes. Authors from China have speculated that qAnti-HBc, being the surrogate marker of the host immune response, could be associated with a better response to transarterial chemoembolisation (TACE) therapy for HCC, developed on cirrhosis terrain [54]. They demonstrated that a baseline qAnti-HBc level above 11.88 S/CO (with sensitivity 90.38% and specificity 71.43%) was related to a higher overall survival rate after TACE therapy, but the mechanism underlying this predictive value of qAnti-HBc still needs to be determined.

Liver transplantation (LT) is an established treatment option for end-stage liver disease induced by HBV infection. Despite the prophylactic application of antiviral drugs and anti-HBV immune-globulins (HBIg), HBV recurrence is a significant risk factor of LT and a marker of poor prognosis. The possibility of re-infection is contingent on residual HBV infection in extrahepatic organs and individual immune response [93]. It was proven that low qAnti-HBc and high qHBsAg at baseline were associated with HBV re-infection after LT [55]. Thus, a higher baseline qAnti-HBc level, representing a solid anti-HBV immune response, was predictive of sustained HBsAg loss after LT.

### 4.8. Prediction of Therapy Success and Relapse after Therapy Discontinuation

The management of CHB currently includes two therapeutic approaches: pegylated interferon (PEG-IFNα) and nucleos(t)ide analogues (NAs). NAs are the most widely chosen option because of their high efficiency, favourable tolerability and oral administration [4,94]. Despite NAs efficient suppression of viral DNA replication, a complete (sterilising) cure is not achievable by this therapy since NAs do not affect the persistence of viral cccDNA in the hepatocytes or the viral DNA integrated into the host genome. Thus, virological, serological, biochemical and histological markers are used to evaluate therapy success and as surrogate endpoints [95]. In contrast to NA-based therapy, the use of PEG-IFNα is limited by the high variability of response and feeble tolerability, but it has a finite duration. Selection of patients eligible for IFN therapy depends on disease activity, HBV genotype, stage of the disease and several virological and serological markers competent to predict the individual response probability [4].

#### 4.8.1. Prediction of Therapy-Induced HBeAg Loss

For NA therapy, one stopping rule that is widely accepted (in different guidelines) is limited only to patients who are HBeAg-positive and able to achieve HBeAg seroconversion and HBV DNA undetectability after consolidation therapy [4,90]. Therapy-induced HBeAg loss also precedes HBsAg loss, a less feasible but more important end-point marker. The predictive power of qAnti-HBc for spontaneous HBeAg loss has already been shown [31,32]. In many studies, a high baseline qAnti-HBc level was associated with higher rates of HBeAg seroconversion in patients treated with both PEG-IFNα and NAs [56,57,58,59,60,61,62]. The same was reported for HIV/HBV co-infected individuals treated with tenofovir-containing antiretroviral therapy and for the children treated with entecavir or PEG-IFNα [23,63]. Some authors showed an independent association of qAnti-HBc and HBeAg seroconversion, while others suggested that the combination of qAnti-HBc results with other parameters (lower level of HBV DNA, HBcrAg) had more convincing predictive value [23,56,58,60]. There are some similarities in the results of different studies that tried to propose cut-off values for qAnti-HBc associated with HBeAg loss. Thus, the suggested level of qAnti-HBc predictive of HBeAg loss in NA-treated patients is >4.37–4.65 log IU/mL, and in patients on PEG-IFNα therapy >3.95–4.4 log IU/mL [56,57,59,60]. Fan et al. demonstrated a baseline level of qAnti-HBc >4.4 log IU/mL to be predictive of HBeAg seroconversion in both treatment protocols with the best combination of sensitivity (53.8%) and specificity (68.3%) [56]. 

The level of qAnti-HBc continuously declined during therapy and could rebound after therapy cessation. Further, patients treated with NAs had a steeper decline in qAnti-HBc level than those treated with PEG-IFNα, probably because IFN could induce a more robust host’s immune response that was reflected in qAnti-HBc [56,58,62].

Historically, ALT level, as a surrogate marker of anti-HBV activity, was used to evaluate therapy success. ALT level of 5xULN was shown to be a strong predictor of HBeAg seroconversion in both NA and IFN therapy [96]. The level of qAnti-HBc was correlated with ALT in many studies but was demonstrated to be a superior predictor of HBeAg loss, possibly due to the non-HBV-specific nature of ALT as an indicator of liver cell damage [58,59]. Moreover, higher baseline qAnti-HBc levels were observed to relate to a higher ALT normalisation rate during IFN therapy. 

The exact mechanism of qAnti-HBc predictive power still needs to be elucidated. The high baseline level of qAnti-HBc reflects the patients’ higher adaptive immune status, which correlates with a better outcome during antiviral therapy. Anti-HBc level indicates the activity of HBcAg-specific B lymphocytes, and B cells also produce cytokines that modulate the activity of CD4+ and CD8+ T cells. The cellular immune response is particularly important for controlling HBV infection [56,97]. 

#### 4.8.2. Prediction of Therapy-Induced HBsAg Loss

Since a complete (sterilising) cure is not achievable by current therapies, the next best-proposed endpoint of anti-HBV therapy is termed “functional cure” and defined as sustained HbsAg loss in addition to undetectable HBV DNA, six months post-treatment [95]. Only a few studies have attempted to explore the predictive power of qAnti-HBc for achieving therapy-induced HbsAg loss [64,65,66,67]. The studies differ among themselves in a serological profile of patients starting therapy, types of therapy and times of qAnti-HBc testing. However, there are some general conclusions regarding the kinetics of qAnti-HBc. HBV core antibody levels tended to become significantly lower during therapy and declined more in NA than in IFN-treated patients. Patients with very low levels of baseline qAnti-HBc were more prone to achieve HbsAg-clearance, which is similar to the setting of spontaneous HbsAg loss [33,34]. 

The predictive effect of qAnti-HBc is here wholly opposite compared to HbeAg-positive patients where baseline qAnti-HBc level was associated with higher rates of HbeAg seroconversion. The explanation relies on two arguments. First, some authors already speculated that qAnti-HBc was a surrogate marker for both the activity of HBV-specific adaptive immune response and the intrahepatic HbcAg load [33]. During the HbeAg-positive phases, qAnti-HBc mainly reflects the strength of the host’s immune response and correlates well with ALT but not HBV DNA. In HbeAg-negative phases, qAnti-HBc is determined by intrahepatic HbcAg and thus viral cccDNA, serum HBV DNA and level of qHBsAg. The second explanation is in different qAnti-HBc antibody-composition between phases of infection [65]. Total qAnti-HBc is composed of both IgM and IgG. High levels of qAnti-HBc IgM can only be found in patients with active liver inflammation and were detected at extremely low levels in inactive HbsAg carriers and in patients who responded to antiviral therapy [64]. Thus, in viral and liver disease suppression states, the qAnti-HBc is composed of only IgG.

In addition, the decline of total-anti-HBc during IFN therapy paralleled that of HBV-DNA but could not differentiate between sustained virological response and virological relapse during follow-up [64]. During NA therapy, the decline of qAnti-HBc provided additional value to monitoring HBV DNA and qHBsAg and, unlike qHBsAg, was HBV genotype independent. Low anti-HBc levels before therapy and older age, and elevated ALT were proven to be predictors of HbsAg loss in NA-treated patients [66].

#### 4.8.3. Prediction of Relapse after Therapy Discontinuation

Based on current guidelines, NA therapy should generally be continued until the occurrence of HbsAg seroclearance because stopping NA treatment before achieving this endpoint is associated with relapse [4,87,90]. However, HbsAg loss is uncommon in NA-treated patients even after many years of treatment, and life-long medication is not without drawbacks, such as accumulating costs and still unknown long-term toxicity. Several new biomarkers, including qAnti-HBc, are currently being investigated to understand their potential in monitoring and guiding clinical decisions regarding discontinuation of therapy and follow-up [8].

For patients who discontinued NA therapy, an end-of-treatment (EOT) qAnti-HBc level was found to be associated with clinical relapse, defined as elevated serum ALT (>2xULN) with HBV DNA level <2000 IU/mL [68,69]. A high EOT level was associated with a lower risk of relapse. In patients with low EOT qAnti-HBc titer (<100 IU/mL), Tseng et al. discovered a tendency of higher risk of clinical relapse but not of virological relapse, defined as an only elevation of HBV DNA (<2000 IU/mL) [68]. It was suggested that qAnti-HBc could be an additional biomarker but that it was unlikely to outperform age and qHBsAg as predictors of off-treatment relapse. The other study identified qAnti-HBc >1000 IU/mL and qHBsAg <100 IU/mL as low-risk markers for clinical relapse after NA cessation [69]. After treatment discontinuation, a substantial increase in qAnti-HBc level was observed in patients with a clinical relapse compared to those without. Thus, off-treatment was inferior to EOT measurement of qAnti-HBc for the prediction of clinical relapse.

Similar results were obtained in studies comprising patients after cessation of IFN therapy [70,71]. Patients who experienced relapse had lower qAnti-HBc levels at baseline and EOT and more significant qAnti-HBc decline during therapy. However, the EOT level was highlighted as better because it reflected the actual immune status of patients who achieved a clinical cure. This is supported by Hou et who reported that qAnti-HBc levels tended to rebound after IFN therapy discontinuation but much less in patients with higher baseline levels [58]. Also, a more substantial relapse predictive value of qAnti-HBc was observed in IFN-treated patients, although it was again proven better if qAnti-HBc was combined with another marker, in this case, anti-HBs [70,71]. A higher risk of relapse was identified in patients with EOT qAnti-HBc level < 2.3386 log IU/mL with the sensitivity of 73.3% and specificity of 71.9% [70]. Surprisingly, in IFN-treated patients, the qAnti-HBc level was found to remain low in patients who had already experienced a relapse.

## 5. Conclusions

The exact mechanism beyond the predictive and diagnostic value of qAnti-HBc level still needs to be elucidated. It is recognised that the level of qAnti-HBc is a surrogate marker of the HBV-specific adaptive immune response activity. However, the activated immune response acts as a double-edged sword since, in an attempt to clear the infection, it causes significant liver injury. All this provides qAnti-HBc with the power to distinguish phases of immune activation during chronic infection and predict consequential risks. Further, in Hbe-negative phases of chronic infection, qAnti-HBc was able to serve as a surrogate marker of the intrahepatic HbcAg load and cccDNA, allowing insight into residual viral activity.

Although an international reference standard has been developed for the quantitation of HBV core antibodies, many currently used methods are only semi-quantitative and demand calibration against the standard to express results in IU/mL. This seriously hinders comparing results from different studies and still prevents the definition of common cut-off values. The availability of various diagnostic kits is not universal but is mainly limited to Asian countries, which is why most studies have comprised only patients infected with genotypes B and C. Despite qAnti-HBc being a viral genotype-independent marker, more international research, including individuals infected with different genotypes, would be necessary. 

Quantitation of HBV core antibodies is a new non-invasive biomarker that can be used in an attempt to solve multiple diagnostic problems. It can provide valuable additional information on viral activity, disease progression and treatment response. However, like other newly utilised biomarkers, qAnti-HBc cannot be relied upon as a single diagnostic test to solve all dilemmas. Based on currently available data, it is evident that its diagnostic and prognostic power can be much improved when combined with other diagnostic biomarkers (HBV DNA, HbeAg, qHBsAg, anti-HBs antibodies). Further research, with broader and more varied patient populations, should be undertaken to determine the actual value of this biomarker.

## Figures and Tables

**Figure 1 viruses-15-00373-f001:**
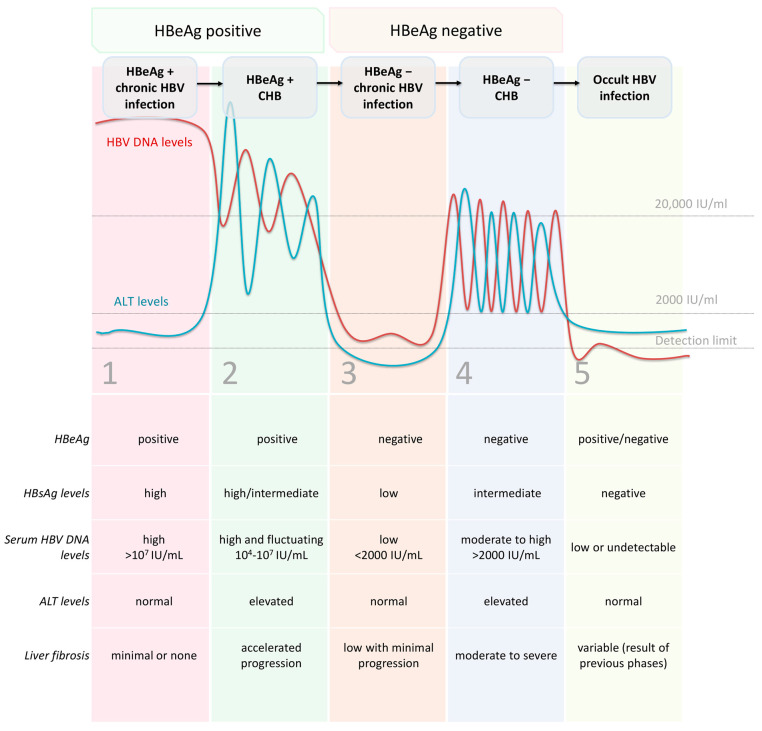
The natural history of chronic HBV infection according to the European Association for the Study of the Liver (EASL). The five phases are not necessarily successive. CHB: chronic hepatitis B.

**Figure 2 viruses-15-00373-f002:**
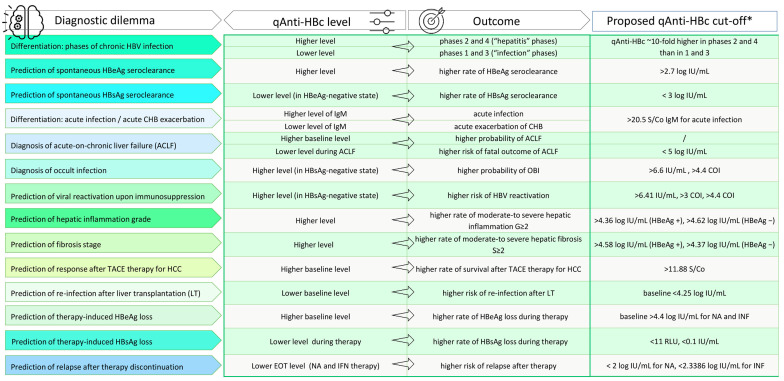
The significant findings on the clinical utility of qAnti-HBc; * the cut-off values vary significantly among studies, one or two are listed for each diagnostic problem; qAnti-HBc: quantitative HBV core antibodies; CHB: chronic hepatitis B; ACLF: acute-on-chronic liver failure; OBI: occult HBV infection; TACE: transarterial chemoembolisation; HCC: hepatocellular carcinoma; LT: liver transplantation; EOT: end-of-treatment; NA: nucleos(t)ide analogue; IFN: interferon; IU: international units; COI: cut-off index; S/Co: sample/cut-off; References associated with diagnostic problems: Refs. [18,26,27,28,29,30] (differentiation of phases of CHB infection), refs. [31,32] (prediction of spontaneous HBeAg seroclearance), refs. [33,34] (prediction of spontaneous HBsAg seroclearance), ref. [35] (differentiation between acute infection and acute CHB exacerbation), ref. [36] (diagnosis of acute-on-chronic liver failure), refs. [20,27,28,37,38] (diagnosis of OBI), refs. [21,37,39,40] (prediction of viral reactivation upon immunosuppression), refs. [19,41,42,43,44,45,46,47,48,49] (prediction of hepatic inflammation grade), refs. [50,51,52,53] (prediction of fibrosis stage), ref. [54] (prediction of response after TACE therapy for HCC), ref. [55] (prediction of re-infection after LT), refs. [23,56,57,58,59,60,61,62,63] (prediction of therapy-induced HBeAg loss), refs. [64,65,66,67] (prediction of therapy-induced HBsAg loss), refs. [68,69,70,71] (prediction of relapse after therapy discontinuation).

## Data Availability

Not applicable.

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
