# Peer review of "Clinical Utility of Quantitative HBV Core Antibodies for Solving Diagnostic Dilemmas"

_viruses, 2023, doi:10.3390/v15020373_

Round 1
Reviewer 1 Report
Thank you very much for giving me this valuable opportunity to review the paper submitted by Lazarevic et al (viruses-2092682). I have reviewed this paper and commented as below:
1- A schematic diagram of the nature history of chronic HBV infection progression should be made according to the statement of Line 40-64.
2- A schematic diagram of methods and units for anti-HBc antibodies quantitation should be made, or a table provided.
3- In Figure 1, would the authors please provide the cuff-off value for higher or lower qAnti-HBc. Maybe, they are different in different studies, if so, would the authors please summarize them in a table.
4- Would the authors please address the sensitivity and specificity for different clinical utility of qAnti-HBc if original publications presented the information.
Author Response
Response to Reviewer 1
Thank you very much for giving me this valuable opportunity to review the paper submitted by Lazarevic et al (viruses-2092682). I have reviewed this paper and commented as below:
1- A schematic diagram of the nature history of chronic HBV infection progression should be made according to the statement of Line 40-64.
A schematic diagram presenting the natural history of chronic HBV infection in accordance with EASL definitions was generated and added to the paper.
2- A schematic diagram of methods and units for anti-HBc antibodies quantitation should be made, or a table provided.
A table showing methods and units for anti-HBc antibodies quantitation was made and added.
3- In Figure 1, would the authors please provide the cuff-off value for higher or lower qAnti-HBc. Maybe, they are different in different studies, if so, would the authors please summarize them in a table.
The available cuff-off values for higher or lower qAnti-HBc for specific diagnostic dilemmas were added to Figure 1 (now Figure 2).
4- Would the authors please address the sensitivity and specificity for different clinical utility of qAnti-HBc if original publications presented the information.
The available data on the sensitivity and specificity of qAnti-HBc antibodies for different clinical utilities were added to the text.

Reviewer 2 Report
Review comments
This review is well written and comprehensive in presenting what is known regarding application of anti-HBc in various diagnostic settings and potential utility during therapy of chronic HBV infection. Some modifications to sections 1 and 2 are suggested below.
Section 1
Line 55: HBeAg negativity with chronic HBV with emergent hepatitis is not universally caused by pre-core or basal core promoter mutations. Please revise.
Line 57: For HBsAg-negative phase, occult HBV infection (where symptoms of hepatitis are caused by HBV replication which is not detectable) should be differentiated from functional cure (where no viral replication is present and liver function is normal).
Line 70-72: HBsAg production is not dependent on cccDNA. Moreover, cccDNA can be present but in a transcriptionally inactive state. These concepts should be presented here as well.
Section 2
Line 82 Some detail on the relationship between HBcAg, HBcrAg and HBeAg antigens should be provided.
Line 83: there is only one nucleocapsid in a hepatitis B virus – delete “inner”.
Author Response
Response to Reviewer 2 Comments
This review is well written and comprehensive in presenting what is known regarding application of anti-HBc in various diagnostic settings and potential utility during therapy of chronic HBV infection. Some modifications to sections 1 and 2 are suggested below.
Section 1
Line 55: HBeAg negativity with chronic HBV with emergent hepatitis is not universally caused by pre-core or basal core promoter mutations. Please revise.
The sentence was revised to specify that mutations were not associated with HBeAg-negativity in all cases.
Line 57: For HBsAg-negative phase, occult HBV infection (where symptoms of hepatitis are caused by HBV replication which is not detectable) should be differentiated from functional cure (where no viral replication is present and liver function is normal).
The sentence was added to emphasise the distinction between occult infection and the state of the functional cure.
Line 70-72: HBsAg production is not dependent on cccDNA. Moreover, cccDNA can be present but in a transcriptionally inactive state. These concepts should be presented here as well.
The sentences were added to discuss the possibility of transcriptionally inactive cccDNA and the full origin of HBsAg.
Section 2
Line 82 Some detail on the relationship between HBcAg, HBcrAg and HBeAg antigens should be provided.
The relationship between HBcAg, HBcrAg and HBeAg is clarified in additional sentences.
Line 83: there is only one nucleocapsid in a hepatitis B virus – delete “inner”.
The word “inner” was deleted.

Reviewer 3 Report
This is a very interesting review focusing on the the main findings regarding the levels of HBV Core Antibodies in the different phases of HBV chronic infection and its diagnostic relevance in different clinical settings. The review is accurate, comprehensive and clearly written. I suggest to the authors to consider the following minor points to improve the text:
Lines 51-52 (4) HBeAg-negative CHB (“immune escape-mutant” phase) active chronic hepatitis.
CHB is not necessary. The authors could change in HBeAg-negative active chronic hepatitis (“immune escape-mutant” phase)
Lines 55-56 HBeAg expression in this phase is reduced or abolished as a result of mutations present in pre-core (Pre-C) and/or basal 56 core promoter (BCP) regions. This is true for most patients in this phase, but not for all of them. Please, specify this concept.
Lines 122-123 The detection range was reported to be 2-5 log IU/mL, and the lower limit of quantitation (LLoQ) was 0.25 IU/mL [19]. This sentence should be revised since if the limit of quantitation is 0.25 IU/mL, it is unclear why the lower limit in the detection range should be 2 log IU/ml (100 IU/ml).
Lines 130-131. The lower limit of detection (LLoD) and LLoQ were 0.8 and 0.5, respectively. Please check, since the LLoD should be 0.5 and the LLoQ should be 0.8.
Figure 1. It would be useful to include the references to the publications, that have highlighted the usefulness of assessing the levels of anti-HBc for answering to the diagnostic dilemmas reported in the figure.
Line 324 The sentence should be changed in “Only in rare cases OBI is the result of an infection with S gene mutants
Line 325 “unrecognisible” needs to be better clarified indicating that mutants can lead to the failure of immunoassays for HBsAg recognition.
Author Response
Response to Reviewer 3 Comments
This is a very interesting review focusing on the the main findings regarding the levels of HBV Core Antibodies in the different phases of HBV chronic infection and its diagnostic relevance in different clinical settings. The review is accurate, comprehensive and clearly written. I suggest to the authors to consider the following minor points to improve the text:
Lines 51-52 (4) HBeAg-negative CHB (“immune escape-mutant” phase) active chronic hepatitis.
CHB is not necessary. The authors could change in HBeAg-negative active chronic hepatitis (“immune escape-mutant” phase)
The name of phase 4 was corrected as suggested.
Lines 55-56 HBeAg expression in this phase is reduced or abolished as a result of mutations present in pre-core (Pre-C) and/or basal 56 core promoter (BCP) regions. This is true for most patients in this phase, but not for all of them. Please, specify this concept.
The sentence was corrected to specify that mutations were not associated with HBeAg-negativity in all cases.
Lines 122-123 The detection range was reported to be 2-5 log IU/mL, and the lower limit of quantitation (LLoQ) was 0.25 IU/mL [19]. This sentence should be revised since if the limit of quantitation is 0.25 IU/mL, it is unclear why the lower limit in the detection range should be 2 log IU/ml (100 IU/ml).
The range of 2-5 log IU/mL was not the overall detection range but a linear detection range. The sentence was corrected accordingly.
Lines 130-131. The lower limit of detection (LLoD) and LLoQ were 0.8 and 0.5, respectively. Please check, since the LLoD should be 0.5 and the LLoQ should be 0.8.
The values for LLoD and LLoQ were replaced, which was now corrected. This mistake was present in the text of the original paper by Caviglia et al. 2020 (ref. 20), but from figure 1 in the paper and the author’s poster presentation of the same results, it was clear that the limit of detection was lower than that of quantitation, as expected.
Figure 1. It would be useful to include the references to the publications, that have highlighted the usefulness of assessing the levels of anti-HBc for answering to the diagnostic dilemmas reported in the figure.
The references associated with the levels of anti-HBc for solving specific diagnostic dilemmas were added to Figure 1 (now Figure 2).
Line 324 The sentence should be changed in “Only in rare cases OBI is the result of an infection with S gene mutants
The sentence was corrected as suggested.
Line 325 “unrecognisible” needs to be better clarified indicating that mutants can lead to the failure of immunoassays for HBsAg recognition.
The term “unrecognizable HBsAg” was replaced with an explanation, and the sentence was split in two to improve readability.

Round 2
Reviewer 1 Report
The authors have revised comprehensively the manuscript according to the reviewer's comments.